# Identification of a De Novo Xq26.2 Microduplication Encompassing *FIRRE* Gene in a Child with Intellectual Disability

**DOI:** 10.3390/diagnostics10121009

**Published:** 2020-11-25

**Authors:** Gianmaria Miolo, Laura Bernardini, Anna Capalbo, Anna Favia, Marina Goldoni, Barbara Pivetta, Giovanni Tessitori, Giuseppe Corona

**Affiliations:** 1Medical Laboratory Department, Genetics Section, Pordenone Hospital, 33170 Pordenone, Italy; barbara.pivetta@asfo.sanita.fvg.it (B.P.); giovanni.tessitori@asfo.sanita.fvg.it (G.T.); 2Medical Oncology and Cancer Prevention Unit, Centro di Riferimento Oncologico di Aviano (CRO), IRCCS, 33081 Aviano, Italy; 3Medical Genetics Unit, Casa Sollievo della Sofferenza IRCCS Foundation, 71013 San Giovanni Rotondo, Italy; l.bernardini@css-mendel.it (L.B.); a.capalbo@css-mendel.it (A.C.); m.goldoni@css-mendel.it (M.G.); 4Department of Pediatrics, Pordenone Hospital, 33170 Pordenone, Italy; anna.favia@asfo.sanita.fvg.it; 5Immunopathology and Cancer Biomarkers Unit, Centro di Riferimento Oncologico di Aviano (CRO), IRCCS, 33081 Aviano, Italy; giuseppe.corona@cro.it

**Keywords:** *FIRRE* gene, intellectual disability, lncRNA, microduplication, chromosomal microarray analysis

## Abstract

Long non-coding RNAs (lncRNAs), defined as transcripts of ≥200 nucleotides not translated into protein, have been involved in a wide range of regulatory functions. Their dysregulations have been associated with diverse pathological conditions such as cancer, schizophrenia, Parkinson’s, Huntington’s, Alzheimer’s diseases and Neurodevelopmental Disorders (NDDs), including autism spectrum disorders (ASDs). We report on the case of a five-year-old child with global developmental delay carrying a de novo microduplication on chromosome Xq26.2 region characterized by a DNA copy-number gain spanning about 147 Kb (chrX:130,813,232-130,960,617; GRCh37/hg19). This small microduplication encompassed the exons 2-12 of the functional intergenic repeating RNA element (*FIRRE*) gene (chrX:130,836,678-130,964,671; GRCh37/hg19) that encodes for a lncRNA involved in the maintenance of chromatin repression. The association of such a genetic alteration with a severe neurodevelopmental delay without clear dysmorphic features and congenital abnormalities indicative of syndromic condition further suggests that small Xq26.2 chromosomal region microduplications containing the *FIRRE* gene may be responsible for clinical phenotypes mainly characterized by structural or functioning neurological impairment.

## 1. Introduction

Intellectual disability (ID) is a neurodevelopmental disorder (NDD) that is often characterized by impaired intellectual and skill ability development. Transcriptional dysregulation of protein-coding genes and alterations in chromatin organization controlling are frequently associated with NDDs [1]. However, only a small fraction (1–2%) of all transcribed mRNAs are translated into proteins [2]. Consequently, the majority of mRNAs consists of non-protein-coding transcripts that are thought to play a key role in cellular biological processes, such as: the regulation of protein-coding gene translation, the control of structural cellular integrity, chromatin remodeling, and overseeing protein localization and degradation [3]. Moreover, it is emerging that non-coding RNAs (ncRNAs) dysregulation may also influence the intellectual development [4,5,6,7,8].

There are several different types of ncRNAs classified according to their DNA loci of biogenesis, distribution, structure, site and mechanism of action. In particular, the long non-coding RNAs (lncRNAs) are expressed in different tissues [9] and they have been reported as being involved in a wide range of regulation functions [5,6,7]. A commonly known lncRNA epigenetic function is the transcriptional inactivation of X chromosome occurring during the female development [10,11]. A specific lncRNA derived from the inactive X chromosome (Xist) exerts a cis-transcriptional silencing of the entire X chromosome. Through X chromosome binding, Xist induces the recruitment of multiple protein complexes, leading to the trimethylation of lysine 27 on histone H3 (H3k27me3) and other epigenetic modifications (e.g., H3k9me2-3, H2Ak119ub and H4k20me1) that favor chromatin packaging, thus blocking the X chromosome transcription [11,12].

Furthermore, the lncRNAs dysregulation have been associated with diverse pathological conditions such as cancer, schizophrenia, Parkinson’s, Huntington’s, Alzheimer’s diseases and NDDs including autism spectrum disorders (ASDs) [6,13,14,15]. Approximately 40% of lncRNAs are distributed in the Central Nervous System (CNS) [8,9], some of which are characterized by highly restricted expression patterns within brain substructures and by different expression during brain development [16,17].

Recently, it has been demonstrated that altered brain levels of lnc-NR2F1 observed in ASD and ID patients may induce changes in the expression of genes involved in neuronal morphological maturation and axon guidance processes that underline the importance of lncRNAs in brain development and functioning [5].

In this report we describe, for the first time, the case of a child carrying a small de novo microduplication of Xq26.2 chromosomal region encompassing the functional intergenic repeating RNA element (*FIRRE*) gene that codes for a lncRNA transcript that could be responsible for an ID phenotype.

## 2. Clinical Report

A 5-year-old child was referred to our clinical Institute for global developmental delay. He was born at 36 weeks for gestational hypertension from unrelated Caucasian parents. Birth weight was 2300 g (0.32 SD); at one month he weighed 2840 g (0.00 SD) with a length of 47 cm (−0.48 SD) and a head circumference of 34.6 cm (0.06 SD). The neonatal and perinatal periods were unremarkable despite the detection of a small atrial septal defect not later detected. At 8 months, he did not sit unsupported; at 14 months he walked with support whereas he spoke his first words at 15 months. At two years he had a single episode of febrile seizure without consequences. Vaccinations were completed on time. Despite regular weight–height development, at 3-and-a-half years, the teachers noticed that the child presented motor defects, hyperactivity and language delay. Therefore, speech therapy sessions began. The child’s pedigree analysis did not detect any case of ID or congenital malformations.

The physical examination performed at 5 years revealed a height of 112 cm (0.49 SD), a weight of 18 kg (0.00 SD), a head circumference of 52 cm (0.18 SD) and a normal-shaped skull with slightly flat occiput. Moreover, he presented a high forehead, normally distributed brown hair, triangular face with tapered chin, normally represented eyelashes and horizontal eyelid fissures. Ears were normal-shaped, with a length of 5.5 cm (0.31 SD). No lobe, helix and antihelix abnormalities or skin appendages or preauricular pits were revealed. Inner canthal distance was 2.7 cm (0.00 SD). The nose was characterized by a depressed bridge, normal root and wings, slightly anteverted nostrils and prominent columella. Filter was smoothed, lower lip was thicker than the upper one and the tongue was normal protruding (Figure 1A,B). Hands’ fingers were apparently short without onychopathies. Palmar folds were regular. No scoliosis problems or flat feet or café au lait spots were detected.

Audiogram examination highlighted a mild bilateral transmission hearing loss with tympanogram type C, whereas the brain Nuclear Magnetic Resonance (NMR) Imaging showed that millimetric hyperintense areoles of non-specific gliosis in the bilateral white substance, mostly in the occipital region, were appreciable. (Figure 2A,B).

## 3. Materials and Methods

Genomic DNA was extracted from the child’s peripheral blood sample using a column-based extraction kit according to the manufacturer’s instructions (NucleoSpin Blood DNA; Macherey-Nagel, Düren, Germany). All the genetic investigations were performed according to the European Guidelines for constitutional cytogenomic analysis [18]. The DNA was investigated by Array-Comparative Genomic Hybridization (Array CGH), according to the manufacturer’s protocol (180K, Agilent Technologies, Walldbronn, Germany) and analyzed by CytoGenomics (v5.0.2.5; Agilent, ADM-2 algorithm, release hg19) only considering the aberrations represented by at least 3 consecutive oligos with a log ratio >±0.25. 

Real-Time quantitative PCR (qPCR) was performed to DNA quantification of *FIRRE* (primers Fw: 5’-TCGCGATGGCAGTAGTGC-3’ and Rv: 5’-GGCTCTGGAACAGTGCTTCG-3’) and *SLC47A2* (primers Fw: 5’-CTTATCAGGGTGCCCAGGAC-3’ and Rv: 5’-CCTTTTGTCTCTTCCAGTTGGC-3’) as target genes, whereas *TERT* gene was considered as reference, using an ABI 7900 Sequence Detection System (Applied Biosystems, Foster City, CA, USA) and DNA-binding dye SYBR Green (Invitrogen Corporation, Carlsbad, CA, USA), as described in Carbone et al. (2008) [19]. For calculation of gene copy-number, we used the 2^−ΔΔCt^ comparative method. All subjects of this study gave their informed consent for inclusion before they participated in the study which was conducted in accordance with the Declaration of Helsinki.

## 4. Results

Standard chromosome analysis using QFQ-banding technique revealed a normal male karyotype 46,XY (550 band resolution). Moreover, molecular analysis of the FMR1 gene detected one allele with 31 (+/−1) CGG repeats in normal range, thus excluding the X-fragile syndrome. 

Array CGH analysis showed a microduplication on the long arm of chromosome X, within region Xq26.2, spanning about 147 Kb (chrX:130,813,232-130,960,617; GRCh37/hg19) (Figure 3a,b). The microduplicated region encompassed the exons 2-12 (Figure 3c) of Online Mendelian Inheritance In Man (OMIM) gene *FIRRE* (*300999) (chrX:130,836,678-130,964,671; GRCh37/hg19) qPCR analysis, extended to parents’ DNA allowed to establish the de novo origin of the microduplication (Figure 3d).

Moreover, the child carried a microduplication on the short arm of chromosome 17, spanning about 222 Kb at 17p11.2 (chr17:19,579,497-19,801,544; hg19) that encompassed the morbid gene *ALDH3A2* (*609523), and further 3 OMIM genes: *ALDH3A1* (*100660), *SLC47A2* (**609833*), and *ULK2* (**608650*). This microduplication was inherited from his healthy father.

## 5. Discussion

In this study we report on a clinical case of a five-year-old child with global developmental delay carrying a de novo microduplication involving the Xq26.2 chromosomal region. This microrearrangement contained the *FIRRE* gene that encodes for a lncRNA involved in the maintenance of chromatin repression through the crosslinking inter-chromosomal interactions with the X chromosome [20].

The *FIRRE* lncRNA has been reported to be involved in multiple processes including adipogenesis, nuclear architecture, haematopoiesis as well as in NDDs [5,17,21,22]. The *FIRRE* gene transcripts contain a unique 156-bp repeating RNA domain (RRD) that allows them to bind to heterogeneous nuclear ribonucleoprotein U (hnRNPU). Then, this latter facilitates the interactions between *FIRRE* transcription locus with other transchromosomal regions leading to the formation of a functional nuclear domain consisting of different genes in close proximity. Therefore, as many other lncRNAs, it was hypothesized to be implicated in the intra- and inter-chromosomal regulation of gene expression and in shaping 3D nuclear organization [20]. The *FIRRE* gene microduplications, as observed in this clinical case, generate lncRNA isoforms characterized by multiple 156-bp RRDs. Such kind of alteration may induce structural changes in transchromosomal compartment able to induce profound perturbation in neuronal cell physiology [23].

In addition to its contribution in three-dimensional topological nuclear organization, *Firre* locus has been implicated in the regulation of gene expression programs in different cellular contexts [20]. Interestingly, the *Firre* locus deletion is accompanied by significant gene expression alterations in mouse hematopoietic progenitor cells, as well as its overexpression increased the levels of cytokines involved in innate immune response, including IL12p40 and TNF-α [17]. However, *Firre* locus does not seem to regulate the local gene expression on the X chromosome through a cis-acting modality, since its alterations are not associated with an increased expression of the neighbouring genes [17]. In agreement with this latter evidence, it has been reported that structural changes of the Barr body, due to *Firre* gene alterations, may lead to a wide transcriptional effect on regulation of autosomal genes involved in processes connected to chromosome structure and segregation [24].

Taken together, these findings seem to demonstrate that *Firre* locus is able, through a trans-acting RNA mechanism, to differently perturb the autosomal gene expression at diverse times of the development and in various tissues, including the brain [17,24].

Different *FIRRE* gene microduplications have been previously reported as being associated with NDDs. In particular, a DNA copy number gain of 0.8 Mb in the Xq26.2 chromosomal region was found in a boy with mild intellectual disability, distinctive facial features, short stature, microcephaly, cardiac disorders and bilateral periventricular nodular heterotopia [21]. Since the same genetic alteration was detected also in the healthy mother, the X-inactivation process may play a pivotal role in maintaining normal neuronal development. However, it is worth noting that the *FIRRE* gene has been reported to escape the epigenetic silencing of chromosome X [20,25], determining an asymmetric expression of two alleles [12,20]. Furthermore, the expression of the *FIRRE* gene is characterized by distinct lncRNA isoforms predominantly derived from active chromosome X (Xa) over the inactive X chromosome (Xi) [12,26]. Therefore, it could be hypothesized that, in the healthy mother of the patient, the altered FIRRE isoforms expression is maintained at a threshold level that does not generate any neurological perturbation.

The patient described in this case report showed a specific phenotype characterized by global developmental delay without additional dysmorphisms and congenital abnormalities indicative of syndromic condition. However, areoles of non-specific gliosis in the bilateral white substance, mostly in the occipital region, were observed via NMR Imaging.

The gliosis process develops from the activation of the microglia cells which play a key role in the regulation of neuronal excitability, synaptic plasticity, neuronal development and in the induction-modulation of inflammation in response to injury [27]. During fetal development, microglia cells contribute to the regulation of neuronal networks, suggesting that their dysregulation might be involved in ASDs and other NDDs [28,29]. Interestingly, in vitro microglia cells’ oxygen-glucose deprivation and reoxygenation (OGD/R) injury promotes the upregulation of *FIRRE* gene that has been associated with the degradation of IkBα, which leads to the NF-kB pathway activation [22]. As a result, several inflammatory genes are induced to trigger the neuroinflammatory process. In our case, an analogous effect could be exerted by the *FIRRE* gene microduplication itself without any specific underlying injury. Indeed, recent findings support the hypothesis that a subclinical inflammatory status coupled with an increased production of oxidant species and a perturbated defense response is involved in NDDs as well as in Rett syndrome [30].

Despite the fact that cerebral expression of lncRNAs is generally lower as compared with protein-coding genes RNAs, FIRRE lncRNA represents an exception, being abundantly transcripted in brain tissue [17]. Thus, the clinical phenotype characterized by an isolated ID form, as observed in our case, may be associated with the higher expression of *FIRRE* gene in the brain relative to all other tissues similarly to what happens with the alterations of protein encoding genes [31]. This is consistent with the DECIPHER database (http:/decipher.sanger.ac.uk/), in which the majority of patients with microduplications affecting the *FIRRE* gene had a phenotype characterized by ID, global developmental delay, and autistic behavior without extra-neurological involvement. 

As a confounding factor of the study results, the child also harboured a second microduplication involving the 17p11.2 chromosomal region that included *ALDH3A2* (*609523) gene encoding for the Aldehyde Dehydrogenase 3 Family Member A2 involved in the fatty acid metabolism and whose biallelic mutations are causative of Sjogren-Larsson Syndrome, the *ALDH3A1* (*100660) gene that encodes for an isozymic form of aldehyde dehydrogenase (ALDH) expressed at high levels in stomach and in cornea, *SLC47A2* (**609833*) gene encoding a multidrug and toxin extrusion protein through urine and bile and *ULK2* (**608650*) gene encoding for a UNC-51-like serine/threonine protein kinase widely expressed in tissues and involved in cell signalling pathways for apoptosis, autophagy and axonal outgrowth. In particular, among these genes, only the *ALDH3A2*, according to an autosomal recessive inheritance, is associated with a disease-phenotype characterized by ID, congenital ichthyosis and spastic diplegia that does not match with the clinical phenotype observed in our patient. Therefore, based on the paternal origin of the microduplication and the lack of clear findings about the NDDs contribution of the genes contained in this microrearrangement, we did not further investigate its potential phenotypic effect.

## 6. Conclusions

Our data seem to suggest that small microduplications on the Xq26.2 chromosomal region including only the *FIRRE* gene are associated with clinical phenotypes mainly characterized by structural or functioning neurological impairment without clear dysmorphic features and congenital abnormalities. However, further cases are required to confirm such a hypothesis.

## Figures and Tables

**Figure 1 diagnostics-10-01009-f001:**
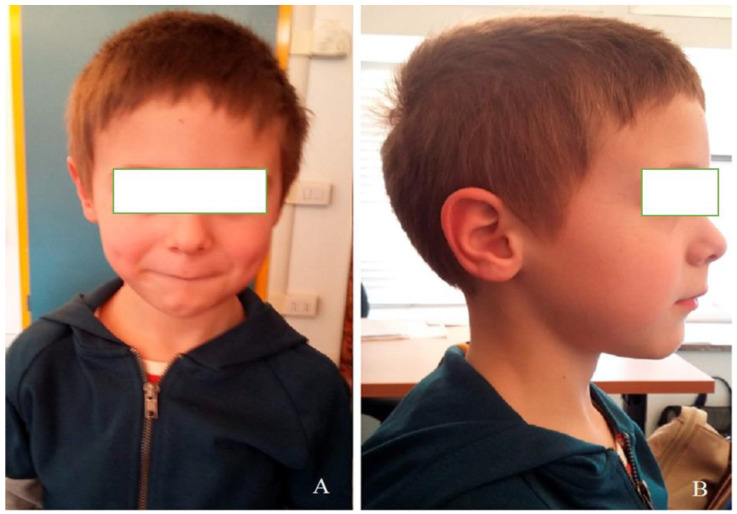
(**A**,**B**) Facial images of the patient at 5 years of age do not show any specific dysmorphisms.

**Figure 2 diagnostics-10-01009-f002:**
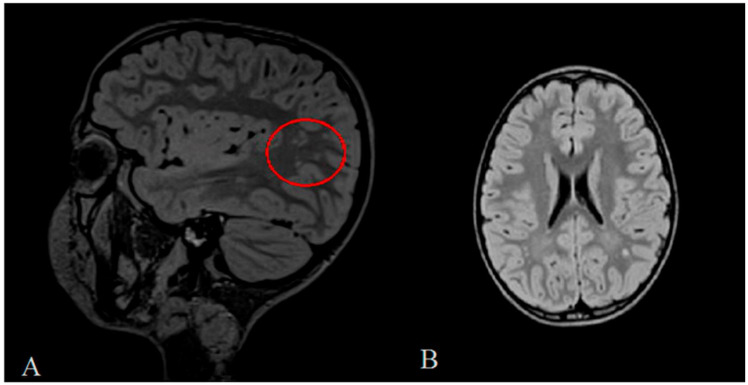
(**A**,**B**) Brain Nuclear Magnetic Resonance (NMR) Imaging showed millimetric hyperintense areoles of non-specific gliosis in the bilateral white substance, mostly in the occipital region as indicated by the red circle.

**Figure 3 diagnostics-10-01009-f003:**
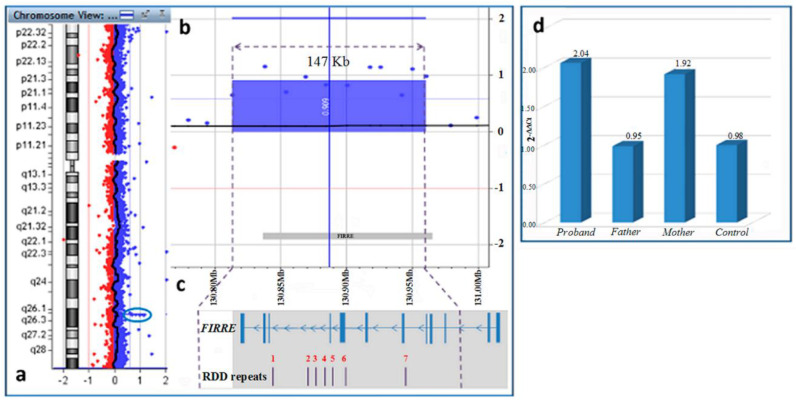
(**a**) genomic profile of chromosome X; (**b**) detailed view of the microduplication of about 147 kb in Xq26.2; (**c**) *FIRRE* exons structure (NR_026975; https://genome.ucsc.edu/) with the localization of 156-bp repeating RNA domains (RRDs), as proposed by Hacisuleyman et al. 2014 [20], all included within the duplicated segment (dashed lines); (**d**) Real-Time quantitative PCR (qPCR) performed to genomic DNA quantification of *FIRRE* gene showed two copies of *FIRRE* (Xq26.2) on the proband’s (duplicated) and maternal (normal) DNA and one copy on the father (normal). A 5-year-old male child was considered as a control.

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
