# Peer review of "Identification of a De Novo Xq26.2 Microduplication Encompassing FIRRE Gene in a Child with Intellectual Disability"

_diagnostics, 2020, doi:10.3390/diagnostics10121009_

Round 1

Reviewer 1 Report

Miolo and colleagues describe a male patient with developmental delay and intellectual disability carrying a 147 kb microduplication encompassing exons 2-12 of the FIRRE gene. The gene encodes a lncRNA involved in the regulation of genes on autosomes. Its abundant level of expression in brain could account for the mainly brain-related clinical manifestations observed in the patient.

This reviewer believes that the following points need to be addressed:

  • The paper requires an extensive revision of the English language
  • References are often inaccurate: for example, the references indicated for rows 49-51 of page 2 are associated with papers where the role of lncRNAs in the contexts of NDDs is discussed. However, the sentence refers to the role of protein-coding genes rather than lncRNAs in NDDs. In rows 54-56 of page 2, the authors describe the physiological functions of non-coding transcripts, but cite papers linking lncRNAs with disease. Other inaccurate references can be found in the rest of the introduction. All the references should be therefore double-checked and adapted.
  • Row 100, page 3: do the authors mean motility or motor defects?
  • Row 139, page 5: city and country of Macherey-Nagel should be mentioned
  • Figure 3d: the qPCR results of the total FIRRE transcript levels are shown. When referring to this figure, the authors should mention age and gender (presumably male) of the control. Additionally, it is already evident from this figure that FIRRE escapes X-inactivation, as observed in the sample of the mother. The authors should comment on that at this stage (in the current version of the manuscript, the escape from X-inactivation is only mentioned in the discussion).
  • An additional microduplication of 17p11.2 is identified in the patient. This microduplication is inherited from the father and includes 4 genes. Only one gene (ALDH3A2) of this microduplication is associated with a disease-phenotype. The authors should underline that this microduplication is most likely not pathogenic not only because of its paternal origin, but also because mode of inheritance (autosomal recessive) and the phenotypical features associated with ALDH3A2 do not match what is observed in the patient.
  • Recent papers (Lewandowski et al., 2019 and Andergassen et al., 2019) prove that FIRRE regulates gene expression on autosomes in an organ-specific manner. Accordingly, the resulting transcriptional dysregulation could be rescued by expression of transgenic FIRRE-RNA. Most neurodevelopmental disorders are caused by mutations in transcriptional regulators or chromatin remodelers, which result in global transcriptional disturbances in the cells. The role played by FIRRE in the context of transcriptional regulation of autosomal genes strengthen the hypothesis that the microduplication observed in the patient might be pathogenic. The authors should expand on that in the discussion.

Reviewer 2 Report

The authors describe a boy with ID, but without dysmorphisms,  and a de novo Xq26.2 duplication, encompassing exons 2-12 of the FIRRE gene. They hypothesize that this duplication could be responsible for the ID phenotype in this patient.

In the clinical report ( 89 -119) there is a detailled description of all the dysmorphisms the patient does NOT have, seems overdone.

In the patient standard chromosome analysis was performed and Array CGH analysis. For persons with ID without dysmorphisms and/or congenital anomalies (so without a distinct phenotype as in this patient) Whole Exome Sequencing (WES) analysis is indicated as the first investigation, because of the high diagnostic yield. In this patient WES was not done, why not? 

The patient should have a WES analysis first to exclude another genetic cause of his ID.

The Array CGH gives a de novo duplication of a part of the FIRRE gene, that has never been reported before. The other patients in the referenced literature with a duplication in Xq26.2 had a larger duplication with more genes in it. There is no reference with a patient with a duplication encompassing only ( a piece of) the FIRRE gene. The possibility that this small duplication of exons 2-12 of the FIRRE gene in the patient the authors describe, is the cause of his ID is therefore limited. Another piece of information that is missing is whether the identified gain region was indeed a tandem duplication of the region or was inserted elsewhere in the genome, in which case a positional effect on another gene would be possible. this could be investigated by fluo-rescence in situ hybridization analysis.

Round 2

Reviewer 2 Report

The authors noticed my remarks and revised the manuscript correctly.